# Cognitive, Affective, and Behavioral Constructs of COVID-19 Health Beliefs: A Comparison Between Sexual Minority and Heterosexual Individuals in Taiwan

**DOI:** 10.3390/ijerph17124282

**Published:** 2020-06-15

**Authors:** Nai-Ying Ko, Wei-Hsin Lu, Yi-Lung Chen, Dian-Jeng Li, Yu-Ping Chang, Peng-Wei Wang, Cheng-Fang Yen

**Affiliations:** 1Department of Nursing, College of Medicine, National Cheng Kung University, Tainan 70101, Taiwan; nyko@mail.ncku.edu.tw; 2Department of Psychiatry, Ditmanson Medical Foundation Chia-Yi Christian Hospital, Chia-Yi City 60002, Taiwan; wiiseen@gmail.com; 3Department of Senior Citizen Service Management, Chia Nan University of Pharmacy and Science, Tainan 71710, Taiwan; 4Department of Healthcare Administration, Asia University, Taichung 41354, Taiwan; elong@asia.edu.tw; 5Department of Psychology, Asia University, Taichung 41354, Taiwan; 6Department of Addiction Science, Kaohsiung Municipal Kai-Syuan Psychiatric Hospital, Kaohsiung 80276, Taiwan; u108800004@kmu.edu.tw; 7Department of Psychiatry, School of Medicine, and Graduate Institute of Medicine, College of Medicine, Kaohsiung Medical University, Kaohsiung 80708, Taiwan; 8School of Nursing, The State University of New York, University at Buffalo, New York, NY 14214-3079, USA; yc73@buffalo.edu; 9Department of Psychiatry, Kaohsiung Medical University Hospital, Kaohsiung 80708, Taiwan

**Keywords:** COVID-19, health belief, LGBT, pandemic, sexual minority

## Abstract

This online survey study aimed to compare the cognitive, affective, and behavioral constructs of health beliefs related to coronavirus disease 2019 (COVID-19) between sexual minority and heterosexual individuals in Taiwan. In total, 533 sexual minority and 1421 heterosexual participants were recruited through a Facebook advertisement. The constructs pertaining to cognition (perceived relative susceptibility to COVID-19, perceived COVID-19 severity, having sufficient knowledge and information on COVID-19, and confidence in coping with COVID-19), affect (worry toward COVID-19), and behavior (adoption of health-protective behaviors) in relation to health beliefs about COVID-19 were compared between sexual minority and heterosexual participants. The results indicated that sexual minority participants had lower perceived susceptibility to COVID-19, greater self-confidence in coping with COVID-19, and lower worry about COVID-19 and were less likely to maintain good indoor ventilation and disinfect their household than heterosexual individuals. Sexual orientation is the modifying factor for the Health Belief Model in the COVID-19 pandemic and should be taken into consideration when medical professionals establish prevention programs for COVID-19.

## 1. Introduction

### 1.1. Coronavirus Disease 2019 Pandemic

Coronavirus disease 2019 (COVID-19) emerged in Wuhan, China, at the end of 2019 and has spread rapidly worldwide; as of 9 June 2020, there have been 7,103,682 confirmed cases and 405,805 deaths [1]. The World Health Organization (WHO) declared the ongoing COVID-19 pandemic as a public health emergency demanding global attention [2]. Taiwan was heavily affected by the 2002–2003 Severe Acute Respiratory Syndrome (SARS), which originated from southern China. Taiwan had the third-highest number of SARS cases globally after China and Hong Kong [3]. This experience with SARS has made many Taiwanese people vigilant against COVID-19. The Taiwanese government has also formulated several public health strategies, including risk communication, hygiene practices, and social distancing since the COVID-19 outbreak to manage Taiwan’s response to global pandemics. However, some individuals will fail to appreciate the necessity of these public health strategies or adapt their everyday behavior to them accordingly. Therefore, pandemic risk communication and behavioral interventions will be improved through a better understanding of the factors related to whether one adopts pandemic protection practices during the COVID-19 pandemic.

### 1.2. Applying the Health Belief Model in the COVID-19 Pandemic

The health belief model (HBM) provides a theoretical framework for understanding the determinants affecting whether a person practices health-promoting behaviors [4]. The HBM proposes multiple constructs that predict engagement in health-related behaviors; these constructs include the perception of one’s susceptibility to a health problem, perception of the seriousness of the health problem, perception of the benefits of and barriers to engagement in a health-promoting behavior that decreases disease risk, and one’s self-efficacy toward successfully performing a health-promoting behavior [4,5]. Accordingly, Liao et al. (2014) proposed a model examining the cognitive, affective, and behavioral measures with regard to the risks of contracting (1) influenza A/H1N1 in 2009 and (2) future respiratory infectious diseases in an epidemic or pandemic, of which COVID-19 is a pandemic [6].

### 1.3. Sexual Orientation as a Moderator in the HBM

The HBM suggests the influence of demographic, psychosocial, and structural variables on the aforementioned HBM constructs [4,7]. However, although sexual orientation is one such variable, its influence on the core HBM constructs has been rarely studied. A previous study found that the HBM construct of “health motivation” was more prominent in lesbian women than in heterosexual women when both practiced breast self-examination [8].

In addition to such empirical evidence, the modifying effect of sexual orientation on the HBM constructs—in the context of the COVID-19 pandemic—must be studied for two reasons. First, sexual minorities, including lesbian, gay, bisexual, pansexual, asexual, and unsure individuals, experience both systemic and individual-level barriers in the healthcare system that diminish their quality of care [9]. In particular, sexual minority individuals may be more likely to delay seeking COVID-19-related medical care. Second, according to the minority stress hypothesis [10], the stigma, prejudice, and discrimination toward sexual minority individuals create a hostile social environment that makes chronic stress and mental health problems more likely in this population [10]. A previous study in Taiwan found that 38% and 32.6% of young adult gay and bisexual men reported experiences of victimization of traditional and cyber sexuality-related bullying, respectively, during childhood and adolescence [11]. Moreover, research in Taiwan has found that victimization of sexuality-related bullying during childhood and adolescence increased the risks of depression [12], anxiety [12], suicide [13], addictive substances [14,15], self-identity confusion [16], and low quality of life [17] in early adulthood. Hatzenbuehler (2009) provided evidence for their psychological mediation hypothesis, which stated that the aforementioned stress negatively affects the cognitive, regulatory, and social mechanisms in minorities [18]. This psychological mediation hypothesis implies that the HBM constructs, in the specific context of COVID-19, may differ between sexual minority and heterosexual individuals.

### 1.4. Aims of This Study

Based on the particularization of the HBM by Liao et al. (2014) to respiratory infectious disease pandemics [6], this online survey study compared constructs pertaining to cognition (perceived relative susceptibility to COVID-19, perceived COVID-19 severity relative to SARS, having sufficient knowledge and information on COVID-19, and confidence in coping with COVID-19), affect (worry about COVID-19), and behavior (adoption of health-protective behaviors) in relation to health beliefs about COVID-19 between sexual minority and heterosexual individuals during the COVID-19 pandemic in Taiwan.

## 2. Methods

### 2.1. Participants

Participants were recruited through a Facebook advertisement from 10 April to 23 April 2020. Online surveys are a promising method for assessing how members of the public general public understand and perceive a fast-moving infectious disease outbreak [19]. Facebook users were eligible for this study if they were ≥20 years old and living in Taiwan. The Facebook advertisement included a headline, main text, pop-up banner, and link to the research questionnaire website. We designed the advertisement to appear in the Facebook users’ news feeds, which are a continually updated list of updates from advertisers and the user’s connections (such as friends and the Facebook groups that they have joined). Our advertisement only targeted users’ news feeds, as opposed to other Facebook advertising locations (e.g., the right column) because news feed advertisements are the most effective in recruiting research participants [20]. We targeted the advertisement to Facebook users by location (Taiwan) and language (Chinese), where Facebook’s advertising algorithm determined which users to show our advertisement to. To ensure that sexual minority individuals were recruited, we also posted the link of the Facebook advertisement to the Facebook pages of three Taiwanese health promotion and counseling centers for lesbian, gay, and bisexual individuals.

This study was approved by the Institutional Review Board (IRB) of Kaohsiung Medical University Hospital (KMUHIRB-EXEMPT(I) 20200011). Because participation was voluntary and survey responses were anonymous, the IRB ruled that this study did not require informed consent. Our study participants were given no incentive for participation. At the beginning of the recruitment flyer on Facebook, we introduced that this online survey aimed to understand respondents’ living experiences during the COVID-19 pandemic, as well as that this anonymous survey protected respondents’ privacy. We also introduced that we provided links to COVID-19 information from the Taiwanese Society of Psychiatry, Kaohsiung Medical University Hospital, and Medical College of National Cheng Kung University for participants to learn more about COVID-19. If the Facebook users agreed to participate in this study after reading the introduction, they could continue reading and responding to the online questionnaires.

### 2.2. Measures

#### 2.2.1. HBM Constructs

We examined four cognitive HBM constructs in the specific context of the COVID-19 pandemic. These constructs were based on the Liao et al.’s particularization of the HBM to respiratory infectious disease pandemics [6]. First, perceived relative susceptibility to COVID-19 was inquired through the question “What do you think are your chances of contracting COVID-19 over the next 1 month compared with others outside your family?” [6]. This question was rated from 1 (no chance of contracting COVID-19) to 7 (guaranteed to contract COVID-19). Low and high perceived susceptibilities were indicated by scores <5 and ≥5, respectively [6]. Second, perceived severity of COVID-19 relative to SARS was inquired through the question “How serious is COVID-19 relative to SARS?” [6]. This question was rated from 1 (COVID-19 is much less serious) to 5 (COVID-19 is much more serious). Low and high perceived severities were indicated by scores <4 and ≥4, respectively [6]. Third, the lack of necessary knowledge and information—considered as a barrier in the HBM [4]—was inquired through the yes/no question “Do you think you have sufficient knowledge and information on COVID-19?” Participants were divided into those who had enough and who did not have enough knowledge and information about COVID-19. Fourth, perceived self-confidence in coping with COVID-19 was inquired through the question “How confident are you that you can cope well with COVID-19?” [21]. This question was rated from 1 (not confident at all) to 5 (very confident). Low and high self-confidence levels were indicated by scores <3 and ≥3, respectively.

Regarding the affective construct, the extent of worry toward COVID-19 was inquired through the question “Please rate how worried you are toward COVID-19” [6]. This question was rated from 1 (very mild) to 10 (very severe). Low and high levels of worry were indicated by scores <6 and ≥6, respectively [6].

The behavioral constructs pertained specifically to whether participants practiced everyday COVID-19 prevention. We posed three questions on whether participants (1) avoided crowded places, (2) maintained good indoor ventilation, and (3) disinfected their household frequently in the past 7 days [6]. Participants who responded “yes, due to COVID-19” were classified as practicing everyday COVID-19 protective behaviors [6].

#### 2.2.2. Demographic Characteristics

Data on sexual orientation (whether heterosexual, homosexual, bisexual, pansexual, asexual, or unsure), gender (female and male), age, and education level were collected. Participants were categorized into sexual minority and heterosexual people. High and low education levels were indicated by university qualifications (or above) and high school qualifications (or below), respectively.

### 2.3. Statistical Analysis

Data analysis was performed using SPSS 22.0 statistical software (SPSS Inc., Chicago, IL, USA). Demographic characteristics and cognitive, affective, and behavioral constructs of health beliefs related to COVID-19 were compared between sexual minority and heterosexual participants using χ^2^ and *t* tests. The associations of sexual orientation with the aforementioned measures were examined using logistic regression analysis. The *p*-value, odds ratio (OR), and 95% confidence interval (CI) were used to indicate significance. A two-tailed *p-*value of <0.05 indicated statistical significance.

## 3. Results

### 3.1. Participant Variables

In total, the data of 1954 participants (533 and 1421 sexual minority and heterosexual individuals, respectively) were analyzed, with 77 of the original 2031 participants excluded due to missing data. Among sexual minority participants, 320 identified as homosexual, 164 identified as bisexual, and 49 identified as pansexual, asexual, or unsure. Table 1 provides a comparison of demographic characteristics and cognitive, affective, and behavioral constructs of COVID-19-related health beliefs between sexual minority and heterosexual participants. According to the results, relative to heterosexual participants, sexual minority participants tended to be male, younger, and more highly educated. Sexual minority participants also tended to be more confident in coping with COVID-19 and less likely to practice everyday COVID-19 prevention (avoiding crowded places, maintaining good indoor ventilation, and disinfecting their household frequently).

### 3.2. Factors Related to the Cognitive Constructs

Table 2 presents the results of factors related to the cognitive constructs. Specifically, after gender, age, and education level were controlled for, sexual minority participants had lower perceived susceptibility and greater self-confidence than did heterosexual participants. By contrast, both groups of participants did not differ with respect to perceived COVID-19 severity relative to SARS and having sufficient knowledge and information on COVID-19.

### 3.3. Factors Related to the Affective and Behavioral Constructs

Table 3 presents the results for the affective and behavioral constructs. Specifically, after demographic characteristics and cognitive constructs were controlled for, sexual minority participants were less likely to worry about COVID-19 than did heterosexual participants. In addition, after demographic characteristics and the cognitive and affective constructs were controlled for, sexual minority participants were less likely to maintain good indoor ventilation and disinfect their household frequently.

## 4. Discussion

### 4.1. Cognitive and Affective Constructs of COVID-19 Health Beliefs

The present study found that sexual minority participants were less likely to perceive themselves as being susceptible to COVID-19. According to the HBM [4], lower perceived susceptibility results in a lower likelihood of practicing pandemic-preventing behavior. A study noted that during the 2009 H1N1 influenza pandemic, people who perceived themselves as having a low risk of infection were less likely to wash their hands and less likely to seek vaccination [22,23]. Sexual minority participants were also less likely to worry about COVID-19, after perceived susceptibility to and confidence in coping with COVID-19 were controlled for. Worrying about COVID-19 is a type of health anxiety. A study noted that a lower level of health anxiety results in less healthy habits [24]; this also applies at the macro level to policy, where low levels of collective health anxiety result in laxer public health strategies for managing epidemics and pandemics.

Information on COVID-19 has been proliferating on traditional and social media since the emergence of the pandemic [25]. Health anxiety has also been increasing worldwide with the spread of the disease [26]. A recent study demonstrated the positive associations of severe depression and anxiety with worry about contracting COVID-19 [27]. A study on the 2009 influenza A/H1N1 pandemic in Hong Kong also found that anticipated, experienced, and current worry specific to influenza A/H1N1 risk was significantly associated with the adoption of health-protective behaviors [6]. However, in our study, sexual minority individuals were more optimistic about their susceptibility to COVID-19. Based on the HBM, research found that structural variables including knowledge about a given disease and prior contact with the disease are the individual factors that can affect perceptions of health-related behaviors [7]. We surmise that the longstanding specter of HIV infection in the sexual minority community may be a structural variable affecting the attitude of sexual minority respondents toward COVID-19. HIV prevention and treatment have progressed since the 1980s; the risk of HIV is still a cause for concern among sexual minority individuals [28]. As proposed in a study, health information pertaining to HIV prevention and treatment strategies is transferable to knowledge on COVID-19 [29]. Because sexual minority individuals may have greater HIV knowledge, they may have higher self-confidence in coping with COVID-19; therefore, they feel less worry. Further research is needed to verify this hypothesis.

### 4.2. Behavioral Constructs of COVID-19 Health Beliefs

Sexual minority participants were also less likely to maintain good indoor ventilation and disinfect their household frequently. Based on the HBM, people may need a cue or trigger that can facilitate engagement in health-promoting behaviors [30]. It is hypothesized that heterosexual respondents may practice everyday COVID-19 prevention more than sexual minority ones because of some external cues such as having more families or friends contracting COVID-19 or receiving more information about COVID-19. However, Taiwan has only 443 cases of contracting COVID-19; moreover, the government of Taiwan held daily press conferences to publicly explain the pandemic situation and offer health education to the public. Therefore, external cues did not account for the difference in maintaining indoor ventilation and disinfecting household between sexual minority and heterosexual respondents. In particular, the difference in maintaining indoor ventilation and disinfecting household persisted after controlling for confidence in coping with COVID-19, perceived susceptibility to COVID-19, and worry about COVID-19. The results suggested that there may be different motivations for adopting health-protective behaviors between sexual minority and heterosexual individuals. Home environment and house layout influence the likelihood of maintaining indoor ventilation; disinfecting household needs equipment such as disinfectants. It raises a question: could it be that sexual minority participants do not have enough resources and equipment to maintain good indoor ventilation and disinfect their household frequently? This warrants further study.

### 4.3. Practice Implications

The results of the present study supported the idea that sexual orientation is the modifying factor for the HBM in the COVID-19 pandemic. Medical and public health professionals should take the modifying role of sexual orientation into consideration when establishing prevention programs for COVID-19. First, in addition to examining what factors are related to lower perceived susceptibility to COVID-19, greater self-confidence in coping with COVID-19, lower worry about COVID-19, and less maintaining of indoor ventilation and disinfecting household in sexual minority individuals, medical and public health professionals should promote adequate knowledge, attitudes, and coping strategies toward COVID-19 in the access points that sexual minority people use to obtain information, such as Facebook, bulletin board systems, and the home pages of health promotion and counseling centers for sexual minority people. Second, inviting popular persons who are friendly toward sexual minority people to promote knowledge, attitudes, and coping strategies toward COVID-19 publicly may produce a marked effect. Third, there may be health promotion and counseling centers providing psychological and HIV consultations via telephone and online for sexual minority people. The governments may cooperate with them to provide COVID-19-related consultation for sexual minority people.

### 4.4. Limitations

Although recruiting participants through Facebook is a promising research method targeting the general public during fast-moving infectious disease outbreaks [19], Facebook users may not be representative of the population. A review of a study that recruited participants through Facebook reported a bias in favor of women, young adults, and people with higher education and incomes [31].

## 5. Conclusions

The present study demonstrated that sexual minority participants had lower perceived susceptibility to COVID-19, greater self-confidence in coping with COVID-19, and lower worry about COVID-19 and were less likely to maintain good indoor ventilation and disinfect their household than heterosexual individuals. In addition to taking the role of sexual orientation into consideration when developing prevention programs for COVID-19, further study is needed to examine the underpinning hypotheses regarding the differences.

## Figures and Tables

**Table 1 ijerph-17-04282-t001:** Comparisons of demographic characteristics and cognitive, affective, and behavioral constructs of health belief related to COVID-19 between sexual minority and heterosexual participants (*N* = 1954).

Variables	Heterosexual (*n* = 1421)	Sexual Minority (*n* = 533)	χ^2^ or *t*	*p*
Gender, *n* (%)				
Female	1026 (72.2)	279 (52.3)	69.644	<0.001
Male	395 (27.8)	254 (47.7)		
Age (years), mean (SD)	39.9 (10.9)	32.5 (8.5)	14.208	<0.001
Education level, *n* (%)				
High (university or above)	1240 (87.3)	496 (93.1)	13.218	<0.001
Low (high school or below)	181 (12.7)	37 (6.9)		
*Cognitive constructs*				
Perceived susceptibility to COVID-19, *n* (%)				
Low	1028 (72.3)	399 (74.9)	1.246	0.264
High	393 (27.7)	134 (25.1)		
Perceived COVID-19 severity relative to SARS, *n* (%)				
Low	417 (29.3)	158 (29.6)	0.017	0.898
High	1004 (70.7)	375 (70.4)		
Having enough knowledge and information about COVID-19				
No	136 (9.6)	55 (10.3)		
Yes	1285 (90.4)	478 (89.7)	0.246	0.260
Confidence in coping with COVID-19, *n* (%)				
Low	219 (15.4)	49 (9.2)	12.665	<0.001
High	1202 (84.6)	484 (90.8)		
*Affective constructs*				
Worry about COVID-19, *n* (%)				
Low	511 (36.0)	215 (40.3)	3.180	0.075
High	910 (64.0)	318 (59.7)		
*Behavioral constructs*				
Avoiding crowded places				
No	235 (16.5)	132 (24.8))	17.202	<0.001
Yes	1186 (83.5)	401 (75.2)		
Maintaining good indoor ventilation				
No	634 (44.6)	313 (58.7)	30.887	<0.001
Yes	787 (55.4)	220 (41.3)		
Disinfecting household frequently				
No	691 (48.6)	323 (60.6)	22.257	<0.001
Yes	730 (51.4)	210 (39.4)		

COVID-19: coronavirus disease 2019; SARS: Severe Acute Respiratory Syndrome; SD: standard deviation.

**Table 2 ijerph-17-04282-t002:** Factors related to cognitive constructs of COVID-19 management.

Variables	Perceived High Susceptibility to COVID-19		Perceived COVID-19 Severity Relative to SARS		Having enough Knowledge and Information about COVID-19		High Confidence in Coping with COVID-19	
	AOR (95% CI)	*p*	AOR (95% CI)	*p*	AOR (95% CI)	*p*	AOR (95% CI)	*p*
Sexual minority ^a^	0.736 (0.574–0.943)	0.015	0.912 (0.732–1.138)	0.415	0.975 (0.679–1.399)	0.890	1.808 (1.276–2.563)	0.001
Males ^b^	1.342 (1.076–1.673)	0.009	0.997 (0.816–1.218)	0.977	1.394 (1.008–1.928)	0.044	1.132 (0.842–1.522)	0.412
Age	0.986 (0.976–0.996)	0.008	1.006 (0.997–1.015)	0.196	0.988 (0.973–1.003)	0.127	1.005 (0.992–1.018)	0.453
Low educational level ^c^	0.758 (0.537–1.070)	0.115	0.973 (0.728–1.301)	0.855	2.013 (1.336–3.034)	0.001	0.878 (0.590–1.308)	0.523

^a^ Heterosexuality as reference; ^b^ Female as reference; ^c^ High school or below as reference. AOR: adjusted odds ratio; COVID-19: coronavirus disease 2019; SARS: Severe Acute Respiratory Syndrome.

**Table 3 ijerph-17-04282-t003:** Factors related to affective and behavioral constructs of COVID-19 management.

Variables	High Worry about COVID-19		Avoiding Crowded Places		Maintaining Good Indoor Ventilation		Disinfecting Household Frequently	
	AOR (95% CI)	*p*	AOR (95% CI)	*p*	AOR (95% CI)	*p*	AOR (95% CI)	*p*
Sexual minority ^a^	0.789 (0.626–0.994)	0.045	0.769 (0.584–1.011)	0.060	0.731 (0.585–0.914)	0.006	0.720 (0.576–0.901)	0.004
Males ^b^	0.642 (0.522–0.791)	<0.001	0.879 (0.679–1.138)	0.328	0.807 (0.657–0.991)	0.041	0.956 (0.779–1.173)	0.667
Age	0.976 (0.967–0.986)	<0.001	1.029 (1.016–1.042)	<0.001	1.024 (1.015–1.034)	<0.001	1.016 (1.007–1.025)	0.001
Low educational level ^c^	0.733 (0.543–0.990)	0.043	0.639 (0.447–0.913)	0.014	1.113 (0.826–1.499)	0.482	1.005 (0.749–1.349)	0.973
High confidence in coping with COVID-19 ^d^	0.307 (0.218–0.433)	<0.001	0.760 (0.507–1.138)	0.182	0.722 (0.548–0.951)	0.021	0.766 (0.585–1.004)	0.053
Perceived high susceptibility to COVID-19 ^e^	1.878 (1.496–2.358)	<0.001	1.034 (0.783–1.366)	0.815	1.143 (0.927–1.409)	0.211	0.938 (0.762–1.155)	0.549
High worry about COVID-19 ^f^			3.323 (2.596–4.255)	<0.001	2.144 (1.759–2.613)	<0.001	2.226 (1.826–2.713)	<0.001

^a^ Heterosexuality as reference; ^b^ Female as reference; ^c^ High school or below as reference; ^d^ Low confidence in coping with COVID-19 as reference; ^e^ Perceived low susceptibility to COVID-19 as reference; ^f^ Low worry about COVID-19 as reference. AOR: adjusted odds ratio

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
