# Peer review of "Cognitive, Affective, and Behavioral Constructs of COVID-19 Health Beliefs: A Comparison Between Sexual Minority and Heterosexual Individuals in Taiwan"

_ijerph, 2020, doi:10.3390/ijerph17124282_

Round 1
Reviewer 1 Report
This is a research paper that compares the cognitive, affective, and behavioural constructs of health beliefs related to coronavirus disease 2019 (COVID-19) between sexual minority and heterosexual individuals in Taiwan. This paper was written with a clear structure and plain language writing style. The issue discussed in this paper is concerning the current COVID-19 pandemic’s effect on different sexual identities. I see how the research topic can be valuable, adding to health behavioural research. However, the content related to the discussion of these findings may need to be reorganized for clarity. For example, the conclusion stating that “sexual minority individuals were more likely to have sleep disturbance and suicidal ideation than heterosexual individuals during COVID-19” is not clearly illustrated from the data provided; especially since the two outcome measures may have been co-existed before the pandemic outbreak (according to your team’s article “suicidality among gay and bisexual men in Taiwan” published in 2018), and may have shown a higher proportion than heterosexual individual before the pandemic outbreak. When seeing COVID-19 pandemic as additive stress onto what sexual minorities have been experienced in Taiwan, this paper would be strengthened if authors could readdress more of your study findings and draw the conclusion of your findings carefully.
Specific comments are as follows:
Introduction
Page 2
- This section was well-written and well-structure. It has a clear theoretical framework (the health belief model) and it also addresses why sexual minorities may bear a higher risk of mental health during this pandemic.
- However, this section will be strengthened if the authors could focus on those literature published in Taiwan. Instead of citing most of your literature from non-Taiwanese countries, studies pertaining to mental health among Taiwanese will provide a powerful standing point to explain why Taiwan needs your study (Taiwanese background information). Otherwise, your readers may lose a wonderful chance to know about mental health issues among individuals in Taiwan.
Methods
Page 3 Line 24-30
- I understand that your study had obtained IRB approval from the Kaohsiung Medical University Hospital in Taiwan. I still have concerns regarding your consent process and resources related to suicidal prevention.
- Most of the online surveys globally were voluntary and anonymous, some of them easily obtained consent by stating “when you forward to the next page, you consent to take part in this study”. Did this study have any other approaches to obtain informed consent from your participants online?
- If you did not obtain informed consent from the beginning of the data collection, the other way to solve this issue is to add on a few sentences, besides from your IRB, to address what information you showed on your recruitment flyer on FB, in order to ensure that your participants did fully understand what potential risks they were going to face, especially one of your questions was related to suicidal ideation.
Page 4
- Please clarify what information you provided when you measured suicidal ideation, such as a list of referral information related to suicide.
- As 10% (n = 206) of your participants showed that they DID have suicidal ideation, it may be unethical to provide no information to this vulnerable population without any incentives.
Results
Page 4
Line 33
- You may want to change “Patient variables” into “Participant variables” since your participants were recruited from an online survey instead of your hospital.
- Or demographic characteristics which you have used on page 8.
Page 5
Table 1
- This table shows your bivariable analyses, however, it is difficult to connect this table with your research topic, especially with the cognitive, affective, and behavioural factors. It would be helpful for your readers to understand your framework if you could reorganize your table 1 using subtitles.
- I also do not understand why all the variables are centred in this table.
- Why “Having enough knowledge and information about COVID-19” was started with the answer “Yes”, when all the others were started with “No”. You may want to make it look consistent.
Page 8.
- After controlling socio-demographic characteristics, the odds ratio should be adjusted odds ratio (AOR) in table 4.
- In table 4, why you decide to remove two factors (maintaining good indoor ventilation & Disinfecting household frequently) from the behavioural constructs that were statistically significance in your bivariate analyses and multivariable analyses? Could you please provide your rationale?
- I feel confused about why you want to include sleep disturbance and suicidal ideation in your study. I would suggest removing sleep disturbance and suicidal ideation from this paper, so this article would be clearly discussed cognitive, affective, and behavioural constructs of health beliefs related to COVID-19 between sexual minority and heterosexual individuals in Taiwan.
Discussion
Page 9
Line 3-9
- I understand that you want to connect lifelong trauma with emotional distance among sexual minorities. However, your findings are not strong enough to explain why sexual minority individuals may have higher optimism about their susceptibility to COVID-19 than heterosexual individuals in Taiwan when you also know that heterosexual individuals did experience bullying in their lifetime.
Line 19-20
- I would encourage the authors to further explain why sexual minority individuals in Taiwan were less likely to maintain indoor ventilation & disinfecting household frequently. Could it be that they do not have enough resources and equipment to do so?
- I would encourage the authors to meaningfully discuss your findings and why there is a difference across sexual orientation. For example, why heterosexual individuals more likely to maintain indoor ventilation & disinfecting household frequently?
Line 26-27
- “…because of increased rates of sleep disturbance and suicidal ideation in sexual minority individuals.” Where did you draw this “increased rates” conclusion from?

Author Response
General comment
Comment 1
The content related to the discussion of these findings may need to be reorganized for clarity. For example, the conclusion stating that “sexual minority individuals were more likely to have sleep disturbance and suicidal ideation than heterosexual individuals during COVID-19” is not clearly illustrated from the data provided; especially since the two outcome measures may have been co-existed before the pandemic outbreak (according to your team’s article “suicidality among gay and bisexual men in Taiwan” published in 2018), and may have shown a higher proportion than heterosexual individual before the pandemic outbreak. When seeing COVID-19 pandemic as additive stress onto what sexual minorities have been experienced in Taiwan, this paper would be strengthened if authors could readdress more of your study findings and draw the conclusion of your findings carefully.
Response
Both reviewers questioned the necessity of examining sleep disturbance and suicidal ideation in this study (Comment 8 of Reviewer 1 and Comment 1 of Reviewer 2). Therefore, we removed the contents regarding sleep disturbance and suicidal ideation from the revised manuscript. We also reorganized the contents of Discussion into “4.1. Cognitive and Affective Constructs of COVID-19 Health Beliefs”, “4.2. Behavioral Constructs of COVID-19 Health Beliefs”, “4.3. Practice Implications’, and “4.4. Limitations.” Please refer to line 208-276.
Specific comments
Comment 2
Introduction
This section will be strengthened if the authors could focus on those literature published in Taiwan. Instead of citing most of your literature from non-Taiwanese countries, studies pertaining to mental health among Taiwanese will provide a powerful standing point to explain why Taiwan needs your study (Taiwanese background information). Otherwise, your readers may lose a wonderful chance to know about mental health issues among individuals in Taiwan.
Response
Thank you for your suggestion. We added 7 references regarding sexuality-related bullying victimization and its effects of mental health among sexual minority people in Taiwan into Introduction section. Please refer to line 81-87.
“A previous study in Taiwan found that 38% and 32.6 of young adult gay and bisexual men reported the experiences of victimization of traditional and cyber sexuality related bullying, respectively during childhood and adolescence [11]. Moreover, research in Taiwan has found that victimization of sexuality related bullying during childhood and adolescence increased the risks of depression [12], anxiety [12], suicide [13], addictive substances [14,15], self-identity confusion [16] and low quality of life [17] in early adulthood.”
Comment 3
Methods
I understand that your study had obtained IRB approval from the Kaohsiung Medical University Hospital in Taiwan. I still have concerns regarding your consent process and resources related to suicidal prevention. Most of the online surveys globally were voluntary and anonymous, some of them easily obtained consent by stating “when you forward to the next page, you consent to take part in this study”. Did this study have any other approaches to obtain informed consent from your participants online? If you did not obtain informed consent from the beginning of the data collection, the other way to solve this issue is to add on a few sentences, besides from your IRB, to address what information you showed on your recruitment flyer on FB, in order to ensure that your participants did fully understand what potential risks they were going to face, especially one of your questions was related to suicidal ideation.
Response
Thank you for your valuable suggestions. We will apply your suggestion in our further online survey. We added the introduction of this online survey at the beginning of the recruitment flyer on Facebook as below into the revised manuscript. Please refer to line 120-127.
“At the beginning of the recruitment flyer on Facebook, we introduced that this online survey aimed to understand respondents’ living experiences during the COVID-19 pandemic, as well as that this anonymous survey protected respondents’ privacy. We also introduced that we provided links to COVID-19 information from the Taiwanese Society of Psychiatry, Kaohsiung Medical University Hospital, and Medical College of National Cheng Kung University for participants to learn more about COVID-19. If the Facebook users agreed to participate in this study after reading the introduction, they could continue reading and responding to the online questionnaires.”
Comment 4
Please clarify what information you provided when you measured suicidal ideation, such as a list of referral information related to suicide. As 10% (n = 206) of your participants showed that they DID have suicidal ideation, it may be unethical to provide no information to this vulnerable population without any incentives.
Response
We agree that providing necessary information for the respondents with suicidal idea is important. Although the revised manuscript removed the contents regarding suicidal ideation, we have provided links at the end of the online questionnaire to the website of the Taiwanese Society of Psychiatry, in which “the Taiwanese Mental Health Guideline for COVID-19 Global Pandemic” was provided for the public to support people with mental ill health (http://www.sop.org.tw/news/l_info.asp?/37.html).
Comment 5
Results
You may want to change “Patient variables” into “Participant variables” since your participants were recruited from an online survey instead of your hospital. Or demographic characteristics which you have used on page 8.
Response
Thank you for your reminding. In the revised manuscript we changed “Patient variables” into “Participant variables.” Please refer to line 170.
Comment 6
Table 1
6.1 This table shows your bivariable analyses, however, it is difficult to connect this table with your research topic, especially with the cognitive, affective, and behavioural factors. It would be helpful for your readers to understand your framework if you could reorganize your table 1 using subtitles.
6.2 I also do not understand why all the variables are centred in this table.
6.3 Why “Having enough knowledge and information about COVID-19” was started with the answer “Yes”, when all the others were started with “No”. You may want to make it look consistent.
Response
6.1 Thank you for your suggestion. We added subtitle “Cognitive constructs”, “Affective constructs” and “Behavioral constructs” into Table 1.
6.2 The journal editors centred the variables in the process of transforming the manuscript into the version for review. We revised them in all three tables of the revised manuscript.
6.3 We revised it by starting with the answer “No” to make it consistent. Please refer to Table 1.
Comment 7
After controlling socio-demographic characteristics, the odds ratio should be adjusted odds ratio (AOR) in table 4.
Response
We changed OR into AOR in Tables 2 and 3.
Comment 8
In table 4, why you decide to remove two factors (maintaining good indoor ventilation & Disinfecting household frequently) from the behavioural constructs that were statistically significance in your bivariate analyses and multivariable analyses? Could you please provide your rationale?
Response
Based on both reviewers’ suggestion, we removed the contents regarding sleep disturbance and suicidal ideation, including Table 4 from the revised manuscript.
Comment 9
I feel confused about why you want to include sleep disturbance and suicidal ideation in your study. I would suggest removing sleep disturbance and suicidal ideation from this paper, so this article would be clearly discussed cognitive, affective, and behavioural constructs of health beliefs related to COVID-19 between sexual minority and heterosexual individuals in Taiwan.
Response
Both reviewers questioned the necessity of examining sleep disturbance and suicidal ideation in this study (Comment 8 of Reviewer 1 and Comment 1 of Reviewer 2). Therefore, we removed the contents regarding sleep disturbance and suicidal ideation from the revised manuscript. We also reorganized the contents of Discussion into “4.1. Cognitive and Affective Constructs of COVID-19 Health Beliefs”, “4.2. Behavioral Constructs of COVID-19 Health Beliefs”, “4.3. Practice Implications’, and “4.4. Limitations.” Please refer to line 208-276.
Comment 10
Discussion
I understand that you want to connect lifelong trauma with emotional distance among sexual minorities. However, your findings are not strong enough to explain why sexual minority individuals may have higher optimism about their susceptibility to COVID-19 than heterosexual individuals in Taiwan when you also know that heterosexual individuals did experience bullying in their lifetime.
Response
Thank you for your comment. We agree that this conjecture did not receive support from our study. We deleted it from the revised manuscript. We also rewrote the paragraph “4.1. Cognitive and Affective Constructs of COVID-19 Health Beliefs” as below. Please refer to line 226-235.
“Based on the HBM, research found that structural variables including knowledge about a given disease and prior contact with the disease are the individual factors that can affect perceptions of health-related behaviors [7]. We surmise that the longstanding specter of HIV infection in the sexual minority community may be a structural variable affecting the attitude of sexual minority respondents toward COVID-19. HIV prevention and treatment have progressed since the 1980s; the risk of HIV is still a cause for concern among sexual minority individuals [28]. As proposed in a study, health information pertaining to HIV prevention and treatment strategies is transferable to knowledge on COVID-19 [29]. Because sexual minority individuals may have greater HIV knowledge, they may have higher self-confidence in coping with COVID-19; therefore, they feeling less worry. Further research is needed to verify this hypothesis.”
Comment 11
I would encourage the authors to further explain why sexual minority individuals in Taiwan were less likely to maintain indoor ventilation & disinfecting household frequently. Could it be that they do not have enough resources and equipment to do so?
Response
Thank you for your suggestion. We rewrote the paragraph “4.2. Behavioral Constructs of COVID-19 Health Beliefs” and added this point into the revised manuscript as below. Please refer to line 238-254.
“Based on the HBM, people may need a cue or trigger that can facilitate engagement in health-promoting behaviors [30]. It is hypothesized that heterosexual respondents may practice everyday COVID-19 prevention more than sexual minority ones because of some external cues such as having more families or friends contracting COVID-19 or receiving more information about COVID-19. However, Taiwan has only 443 cases of contracting COVID-19; moreover, the government of Taiwan held daily press conferences to publicly explain the pandemic situation and offer health education to the public. Therefore, external cues did not account for the difference in maintaining indoor ventilation and disinfecting household between sexual minority and heterosexual respondents. Especially, the difference in maintaining indoor ventilation and disinfecting household persisted after controlling for confidence in coping with COVID-19, perceived susceptibility to COVID-19, and worry about COVID-19. The results suggested that there may be different motivations for adopting health protective behaviors between sexual minority and heterosexual individuals. Home environment and house layout influence the likelihood of maintaining indoor ventilation; disinfecting household needs equipments such as disinfectants. It raises a question: could it be that sexual minority participants do not have enough resources and equipment to maintain good indoor ventilation and disinfect their household frequently? It warrants further study.”
Comment 12
“…because of increased rates of sleep disturbance and suicidal ideation in sexual minority individuals.” Where did you draw this “increased rates” conclusion from?
Response
We removed the contents regarding sleep disturbance and suicidal ideation from the revised manuscript.
Reviewer 2 Report
This article seeks to address the importance of sexual orientation on affecting people practices health-promoting behaviors during the COVID-19 pandemic in Taiwan based on the health belied model (HBM). It is interesting, well-designed, and well-written, and provides a new perspective to know more about the influences caused by the COVID-19.
After reading the article, I would like to invite authors to provide more information on the four issues below:
1. The health belied model (HBM) was used as the theoretical basis for this article. The HBM mainly “provides a theoretical framework for understanding the determinants affecting whether a person practice health-promoting behaviors” (p.2), but does not explain the relationship between people's practice of health-promoting behavior and their mental health. Though the authors cite the Hatzenbuehler’s (2009) psychological adjustment hypothesis, they do not explain clearly the necessity of including the psychological adjustment hypothesis into the HBM. Therefore, it would be helpful if the authors can provide more information on this issue.
2. In the discussion section, the authors mention that “sexual minority individuals were more optimistic about their susceptibility to COVID-19. Besides the possible reasons suggested by the authors, it would be helpful if the authors can provide the possible reasons based on the HBM. In addition, it would be great if the authors can explain the relationship between people's practice of health-promoting behavior and their mental health based on their research results.
3. In the “practice implication” section (p.9), the authors suggest that the modifying role of sexual orientation should be included into the consideration when establishing prevention programs for COVID-19. I would like to invite the authors to provide more specific suggestions for medical and public health professionals. For example, how medical and public health professionals can design appropriate behavioral intervention programs for sexual minority individuals to maintain good indoor ventilation or disinfect household frequently based on the HBM? Furthermore, since the modifying role of sexual orientation should be considered in the behavioral intervention programs for sexual minority individuals, It would be good if the authors can try to explain the relationship between people's practice of health-promoting behavior and their mental health based on the HBM.
4. In the “limitation” section, authors said that their study was conducted during the period of COVID-19 mitigation but not during the period when COVID-19 first emerged in Taiwan. Because the HBM does not include the environmental factors to understand the determinants affecting whether a person practice health-promoting behaviors, it is hard for me to know the potential limitation between the two periods mentioned in this section.
“Patient variables” on p.4 should be “participant variables”.
Author Response
Comment 1
The health belied model (HBM) was used as the theoretical basis for this article. The HBM mainly “provides a theoretical framework for understanding the determinants affecting whether a person practice health-promoting behaviors” (p.2), but does not explain the relationship between people's practice of health-promoting behavior and their mental health. Though the authors cite the Hatzenbuehler’s (2009) psychological adjustment hypothesis, they do not explain clearly the necessity of including the psychological adjustment hypothesis into the HBM. Therefore, it would be helpful if the authors can provide more information on this issue.
Response
Both reviewers questioned the necessity of examining sleep disturbance and suicidal ideation in this study (Comment 8 of Reviewer 1 and Comment 1 of Reviewer 2). Therefore, we removed the contents regarding sleep disturbance and suicidal ideation from the revised manuscript. We are writing a new manuscript focusing on the relationship between people's practice of health-promoting behavior and their mental health and the moderating effect of sexual orientation.
Comment 2-1
In the discussion section, the authors mention that “sexual minority individuals were more optimistic about their susceptibility to COVID-19. Besides the possible reasons suggested by the authors, it would be helpful if the authors can provide the possible reasons based on the HBM.
Response
Thank you for your comment. We rewrote the paragraph “4.1. Cognitive and Affective Constructs of COVID-19 Health Beliefs” as below. Please refer to line 226-235.
“Based on the HBM, research found that structural variables including knowledge about a given disease and prior contact with the disease are the individual factors that can affect perceptions of health-related behaviors [7]. We surmise that the longstanding specter of HIV infection in the sexual minority community may be a structural variable affecting the attitude of sexual minority respondents toward COVID-19. HIV prevention and treatment have progressed since the 1980s; the risk of HIV is still a cause for concern among sexual minority individuals [28]. As proposed in a study, health information pertaining to HIV prevention and treatment strategies is transferable to knowledge on COVID-19 [29]. Because sexual minority individuals may have greater HIV knowledge, they may have higher self-confidence in coping with COVID-19; therefore, they feeling less worry. Further research is needed to verify this hypothesis.”
Comment 2-2
In addition, it would be great if the authors can explain the relationship between people's practice of health-promoting behavior and their mental health based on their research results.
Response
Thank you for your suggestion. We rewrote the paragraph “4.2. Behavioral Constructs of COVID-19 Health Beliefs” and added this point into the revised manuscript as below. Please refer to line 238-254.
“Based on the HBM, people may need a cue or trigger that can facilitate engagement in health-promoting behaviors [30]. It is hypothesized that heterosexual respondents may practice everyday COVID-19 prevention more than sexual minority ones because of some external cues such as having more families or friends contracting COVID-19 or receiving more information about COVID-19. However, Taiwan has only 443 cases of contracting COVID-19; moreover, the government of Taiwan held daily press conferences to publicly explain the pandemic situation and offer health education to the public. Therefore, external cues did not account for the difference in maintaining indoor ventilation and disinfecting household between sexual minority and heterosexual respondents. Especially, the difference in maintaining indoor ventilation and disinfecting household persisted after controlling for confidence in coping with COVID-19, perceived susceptibility to COVID-19, and worry about COVID-19. The results suggested that there may be different motivations for adopting health protective behaviors between sexual minority and heterosexual individuals. Home environment and house layout influence the likelihood of maintaining indoor ventilation; disinfecting household needs equipments such as disinfectants. It raises a question: could it be that sexual minority participants do not have enough resources and equipment to maintain good indoor ventilation and disinfect their household frequently? It warrants further study.”
Comment 3
In the “practice implication” section (p.9), the authors suggest that the modifying role of sexual orientation should be included into the consideration when establishing prevention programs for COVID-19. I would like to invite the authors to provide more specific suggestions for medical and public health professionals. For example, how medical and public health professionals can design appropriate behavioral intervention programs for sexual minority individuals to maintain good indoor ventilation or disinfect household frequently based on the HBM?
Response
Thank you for your suggestion. We rewrote the paragraph “4.3. Practice Implications” and add the contents as below. Please refer to line 259-270.
“First, in addition to examine what factors related to lower perceived susceptibility to COVID-19, greater self-confidence in coping with COVID-19, lower worry about COVID-19, and less maintaining indoor ventilation and disinfecting household in sexual minority individuals, medical and public health professionals should promote adequate knowledge, attitudes and coping strategies toward COVID-19 in the accesses from which sexual minority people used to obtain information such as Facebook, the bulletin board system, and the home pages of health promotion and counseling centers for sexual minority people. Second, inviting the popular persons who are friendly toward sexual minority people to promote knowledge, attitudes and coping strategies toward COVID-19 publicly may produce a marked effect. Third, there may be health promotion and counseling centers providing psychological and HIV consultations via telephone and online for sexual minority people. The governments may cooperate with them to provide COVID-19-related consultation for sexual minority people.”
Comment 4
In the “limitation” section, authors said that their study was conducted during the period of COVID-19 mitigation but not during the period when COVID-19 first emerged in Taiwan. Because the HBM does not include the environmental factors to understand the determinants affecting whether a person practice health-promoting behaviors, it is hard for me to know the potential limitation between the two periods mentioned in this section.
Response
Thank you for your comment. We deleted it from the revise manuscript. Please refer to line 276.
Comment 5
“Patient variables” on p.4 should be “participant variables”.
Response
Thank you for your reminding. In the revised manuscript we changed “Patient variables” into “Participant variables.” Please refer to line 170.
Round 2
Reviewer 1 Report
The authors have addressed my previous comments appropriately and revised the manuscript accordingly. I have no further comments for this manuscript.